# Imidazolium Salt for Enhanced Interfacial Shear Strength in Polyphenylene Sulfide/Ex-PAN Carbon Fiber Composites

**DOI:** 10.3390/polym14173692

**Published:** 2022-09-05

**Authors:** Baptiste Gaumond, Sébastien Livi, Jean-François Gérard, Jannick Duchet-Rumeau

**Affiliations:** Univ Lyon, CNRS, UMR 5223, Ingénierie des Matériaux Polymères, Université Claude Bernard Lyon 1, INSA Lyon, Université Jean Monnet, CEDEX, F-69621 Villeurbanne, France

**Keywords:** imidazolium salt, interphases, thermoplastics, composites, micromechanical tests

## Abstract

Processing structural or semi-structural thermoplastic-based composites is a promising solution to solve the environmental issues of the aeronautic industry. However, these composites must withstand high standard specification to ensure safety during transportation. For this reason, there is a real need to develop strong interactions between thermoplastic polymers and reinforcement fibers. This paper investigates relationships between the surface chemistry, microstructure and micromechanical properties between polyphenylene sulfide and ex-PAN carbon fibers. The incorporation of ionic salt such as 1,3-Bis(4-carboxyphenyl)imidazolium chloride into neat polyphenylene sulfide was able to significantly increase the interfacial shear strength measured by microbond micromechanical test combined with different carbon fiber surfaces treatment.

## 1. Introduction

Fiber reinforced polymers have found a strong place into aerospace applications. These materials disclose low densities while keeping high mechanical properties which makes them essential to the development of the new aircraft generation. These materials help the aeronautical industry to face environmental issue by decreasing the weight and the consumption of the planes. However, there is still environmental issues regarding the end-of-life of these materials which are mainly thermoset composites. Nowadays, thermoplastic composites could be a major alternative to conventional thermoset-based ones used for structural applications. In fact, high performance thermoplastic matrices can combine high mechanical properties including impact resistance, flame retardancy, and solvent resistance. Unlike thermosets, they also respond to environmental issues because of their ability to be stored at room temperature, their high recyclability and their short production process. Among them, Polyphenylene sulfide (PPS) is one of the most widely used high performance thermoplastic polymers with poly(ether ether ketone) (PEEK), poly(ether ketone ketone) (PEKK) and poly(etherimide) (PEI). This polymer is semicrystalline and composed of phenyl rings and sulfur atoms which ensures it its high thermal and mechanical properties. PPS has the specificity to be highly resistant to dissolution in the most common solvents [1] and to possess a good fire retardancy. However, these outstanding performances require high processing temperatures not compatible with conventional carbon fibers sizing and that can lead to a poor interfacial adhesion. The usual methods for improving interfacial adhesion between fibers and polymers include carbon fiber surface modification which can act on mechanical or chemical adhesion [2]. Different routes are described in the literature by using plasma treatments [3,4], chemical treatments [5], nanoparticle coating [6] or compatibilizers [7,8,9]. It is well known that the use of compatibilizers can improve the interfacial adhesion between fiber and matrix by the bridging effect. Moreover compatibilizers could either interact chemically or physically with the fiber surface: (i) it can be deposited onto the fibers or (ii) blended into the matrix [10].

Recently, organic salts denoted “ionic liquids” were reported as excellent multifunctional additives of polymer matrices for processing advanced materials. They can be used as plasticizers, (nano)structuring agents, surfactants or as compatibilizing agent of polymer blends composed of thermoplastics, thermosets or a mixture of thermoset/thermoplastic [11]. Since the use of ionic liquids or organic salts with polymer for multiple applications grow rapidly, it is relevant to characterize their potential influence on the interface/interphase in the organic/inorganic hybrid materials. Some works can be found in the literature. In the work of Donato et al., ionic liquid 1-decyl-3-methylimidazolium tetrafluoroborate was used as coupling agent between silica and polypropylene [12]. Their results show the potential application of ionic liquid into composite materials with better dispersion of inorganic charges into the polymer matrix. Eyckens et al. have recently used ionic liquid (IL) as sizing agents of carbon fibers in order to enhance the interfacial shear strength with both thermoplastic maleic anhydride grafted PP (MAH-PP) and epoxy matrix [13]. Their objective was to plasticize the matrix surrounding the carbon fibers increasing energy dissipation mechanism under shear force. Their works showed a significant increase of the interfacial shear strength (IFSS) determined by fragmentation test between IL sized carbon fibers and epoxy matrix. They highlighted similar results on IFSS measured by pull-out tests between IL sized carbon fibers and MAH-PP. The addition of butyl methyl imidazolium chloride onto the fiber’s surface increased the adhesion strength by 66% compared to the unmodified samples. More recently, the same research group investigated the influence of the impregnation of an ex-Pan carbon fiber with an aryl diazonium salt on the adhesion with epoxy matrix [14]. This work has shown that the presence of this coupling agent at the interphase was able to strongly modify the fiber-matrix adhesion. The authors reported an increase of 300% of the IFSS measured between electrochemically modified (54.0 MPa) and unsized carbon fibers (17.8 MPa) embedded into an epoxy resin. They suggested that this result could be partially attributed to dragging effects between the sizing and the polymer matrix due to non-covalent interactions.

In this work, the objective was to introduce an organic salt based on imidazolium cation directly into the matrix and to investigate its effect on adhesion. However, the dedicated salt must fulfill some requirements as high thermal stability up to 300 °C and the presence of functional group to increase the fiber/matrix compatibility. Thus, we have measured the strength of the interphase built between three ex-PAN carbon fibers with different surface reactivity and polyphenylene sulfide with and without an imidazolium salt, i.e., PPS vs. IL modified PPS. This is the first time that this difunctional imidazolium salt is mixed within the PPS thermoplastic by means of conventional plastic processing tools. This proof of concept must pave the way to new thermally stable compatibilizers able to migrate from the matrix towards the interface to strengthen it efficiently. 

## 2. Experimental Part

### 2.1. Materials

High molecular weight polyphenylene sulfide (PPS) was obtained from Celanese (M_w_ = 56000 g·mol^−1^ Fortron^®^ 0320, Franfurt, Germany)). Epoxy sized (S-CF) and non-sized high resistance carbon fibers (NS-CF) were obtained from Mitsubishi chemical carbon fibers and composites (Irvine, USA) under the commercial name Grafil^®^ 34-700. Non-treated AS4 carbon fibers, i.e., non-sized and non-oxidized (NT-CF) were gracefully provided by Hexcel Composites (Dagneux, France). The salt 1,3-Bis(4-carboxyphenyl)imidazolium Chloride denoted ImCl was synthesized according to the following procedure with some modification (see Figure 1) [15].

To a solution of 4-Aminobenzoic acid (5.00 g, 36.5 mmol, 2.0 eq.) in dry methanol (20 mL), formic acid (2 drops) was added followed by dropwise addition of a 39% aqueous solution of glyoxal (1.99 mL, 17.40 mmol, 1.0 eq.). The solution was stirred at room temperature during a 24 h period. The white solid obtained was collected by filtration, washed with cold methanol, and dried in air. The product *N*,*N*′-Bis(4-carboxyphenyl)ethylenediimine was obtained as a white solid (3.93 g, 79%) and used for the next step without any purification. To a solution of *N*,*N*′-Bis(4-carboxyphenyl)ethylenediimine (3.93 g, 13.3 mmol, 1.0 eq.) in anhydrous THF (11 mL) under an argon atmosphere was added paraformaldehyde (498 mg, 16.6 mmol, 1.25 eq.) and 12 N HCl (1.65 mL, 20.00 mmol, 1.5 eq.) in dioxane (3.2 mL) at 0 °C. The reaction mixture was stirred at room temperature for 4 h. The precipitate formed was collected by filtration, washed with Et_2_O, and dried in vacuum. The product was obtained as a yellow solid (3.09 g, 75%).

^1^H NMR (500 MHz, DMSO-d6) δ 13.42 (bs, 2H), 10.64 (s, 1H), 8.71 (d, J = 1.5 Hz, 2H), 8.24 (d, J = 8.7 Hz, 4H), 8.10 (d, J = 8.7 Hz, 4H).13C NMR (126 MHz, DMSO-d6) δ 166.2, 137.7, 135.5, 132.1, 131.2, 122.1, 121.9. IR (neat) cm^−1^ 3124, 2974, 2810, 2599, 2456, 1704, 1556, 1392, 1231, 1175. Mp: > 400 °C. HRMS m/z (ESI): calcd. for C17H13N2O4 [M]+: 309.0875, found: 309.0878.

Chemical structures of PPS and Imidazolium salt are shown in Table 1.

### 2.2. Processing of PPS and Salt Modified PPS Blends

#### 2.2.1. Polymer Films

A twin-screw DSM microcompounder (Midi 2000, Heerlen, The Netherlands) was used to prepare the neat PPS and the salt modified PPS (PPS/ImCl). The imidazolium salt rate, 5 wt% was chosen because previous works showed that a low rate of compatibilizer is widely enough to modify the interface without plasticizing the matrix or exsudating at the film surface [16]. The mixture was sheared for about 2 min with a 100 rpm speed at 310 °C under a nitrogen flow rate. We used a flat die to obtain 200 μm thickness PPS and PPS_ImCl films required for the microcomposite sample fabrication. 

#### 2.2.2. Microcomposite Processing

Microcomposite samples were obtained using the procedure described in Figure 1. Small parts of polymer films are first cut in the middle to obtain “pants”. These parts are then suspended onto horizontal fibers glued on small cardboard with epoxy resin and melt in an oven at 330 °C for 3 min. The samples are air-quenched at room temperature. The embedded length of fiber within the matrix droplet is then measured by optical microscopy. All carbon fibers diameters are measured by acoustic emission prior to microcomposite specimen fabrication.

#### 2.2.3. Thermal Treatments of Microcomposites

Knowing that PPS is a semicrystalline polymer, the composite properties are clearly affected by its crystallinity which depends on the thermal history and cooling process [17,18,19]. Many reports in the literature investigated the impact of the cooling process on adhesion measurements at the micromechanical scale [20,21,22,23,24]. In these conditions, we decided to perform annealing treatment before the adhesion measurements. The applied annealing treatment (160 °C for 1 h under air atmosphere) allowing to complete the crystallization process was shown in Figure 2. When polymer drops are cooled at the exit of the oven, amorphous microcomposites are distinguishable by their translucence under the microscope (Figure 2a) whereas after applying annealing treatment, a crystalline morphology induces light scattering (Figure 2b).

### 2.3. Characterization Methods

*Fourier Transform Infrared Absorption spectra (FTIR)* were recorded on a Spectrum two FT-IR apparatus (Perkin Elmer, Akron, OH, USA) equipped with a GladiATR controlled in temperature (Piketech, Akron, OH, USA) in transmission mode from 4000 to 500 cm^−1^ (32 scans; resolution 4 cm^−1^). 

*The thermogravimetric analyses* were carried out using a Q500 thermogravimetric analyzer (TA Co., Ltd., New Castle, DE, USA) from ambient to 700 °C at a heating rate of 10 °C·min^−1^ under air atmosphere with a flow rate of 40 mL·min^−1^. 

*Differential Scanning Calorimetry* thermograms of PPS and PPS/ImCl were recorded using a DSC Q20 (TA Co., Ltd., New Castle, DE, USA) under nitrogen flow of 50 mL·min^−1^. The heating and cooling rates were 10 °C·min^−1^.

*Transmission electron microscopy* was performed at the Technical Center of Microstructures (University of Lyon) using a Phillips CM 120 microscope (Waltham, MA, USA) operating at 80 kV to characterize the crystalline morphology. The 60 nm-thick ultrathin sections were obtained using an ultramicrotome equipped with a diamond knife and were then set on copper grids.

*Surface energies* were determined on matrix and on fibers. The contact angle (CA) was deposited on matrix films from the sessile drop method using a OCA15 (Dataphysics, Germany) goniometer. The surface energy on carbon fibers was determined by Wilhelmy method on DCAT21 tensiometer from Dataphysics (Germany). Water, methylene diiodide, ethylene glycol were used as probe liquids according to the Owens–Wendt method to measure non dispersive and dispersive components of surface energy.

*The calculated diameter* of the carbon fibers was determined using a FAVIMAT+ (Textechno, Mönchengladbach, Germany). The apparatus is using the Mersenn’s law to determine the linear density:(1)$$f=\frac{1}{2L}\sqrt{\frac{T}{\mu }}$$
where *f* [Hz] is the resonance frequency, *T* [N] is the tension applied on the fiber, *L* [km] is the sample length of the evaluated carbon fiber and *μ* ([kg·km^−1^]) is the linear density. Knowing the linear density and the volume weight *ρ* [g·cm^−3^] of the fibers, it is then possible to calculate the diameter *d* [µm] considering that fibers are cylinders. According to the manufacturer we used the following parameters for the linear density determination: T = 2 cN·dtex^−1^, L = 50 mm. The volume weight of the carbon fibers used in this study was *ρ* = 1.8 g·cm^−3^.

Further information concerning the operating system and theory can be found in [25].

The observation of microdroplets deposited on carbon fibers was performed using a Zeiss *optical microscope* in reflection mode.

*Scanning electron microscopy (SEM)* was used to observe the carbon fibers surfaces (Merlin compact, Zeiss, Jena, Germany). This microscope is equipped with a traditional detector and an in-lens detector located inside the electron column of the microscope sensitive to chemical contrast.

*Microbond tests* were performed using an MTS 2/M tensile machine (MTS Systems SAS, Creteil, France) equipped with a 10 N force cell. The system used to perform the microbond test consisted in two cutter blades mounted on a movable block that were controlled with micrometer screws. A macroscopic camera was used to help positioning the cutter blades and to monitor the test. The displacement rate was set to 0.1 mm/min. The fiber was pulled-out of the droplet blocked by the cutter blades and the load-displacement was recorded for each test until debonding occurred. The maximum force at debonding, F_max_, was considered together with the interfacial area in order to calculate the interfacial shear stress, as shown in Equation (1) i.e., considering a constant shear stress along the embedded length. Forty single tests were performed for each couple fiber/matrix in order to obtain an averaged IFSS. The debonding zone was observed using an optical microscope and a scanning electron microscope (SEM) to verify the validity of the test: no residual resin must be left behind the fiber.
(2)$$\tau =\frac{F_{\max}}{{\pi L}_{e}d_{f}}$$
where F_max_ is the maximum force, L_e_ is the embedded length and d_f_ is the carbon filament diameter.

## 3. Results and Discussion

### 3.1. Behavior of Pps and Pps_ImCl

#### 3.1.1. Thermal Stability of Pps and Pps_ImCl Matrices

The influence of 1,3-Bis(4-carboxyphenyl)imidazolium Chloride (ImCl) on PPS thermal stability was studied by thermogravimetric analysis (TGA). The evolution of the weight loss as well as the derivative weight loss as a function of the temperature of ImCl, neat PPS, and PPS_ImCl are presented in Figure 3.

In addition, the temperatures corresponding to the first and second peaks of degradation as well as the temperature corresponding to 5 wt% loss are reported in Table 2. The thermal degradation of the imidazolium salt shows three main regions: the first degradation peak is measured at 349 °C, the second one at 435 °C and the last one at 600 °C. 

This ionic compound withstands high processing temperatures thanks to the presence of the imidazolium and the linked aromatic cycles and the presence of the chloride counter anion as discussed in the work of Awad et al. [26]. However, in the case of PPS, we can see that the T_d5%_ measured for ImCl is close to the temperature of extrusion we used for processing our films and extrudates, i.e., Table 2. This result suggests a possible degradation of ImCl during the extrusion. In summary, the presence of the imidazolium salt does not improve the thermal stability of PPS as it can be found in the literature for other polymers [27,28]. It is found that PPS is slightly more stable than the blend PPS_ImCl since a small degradation corresponding to the imidazolium salt happens above 400 °C for the compatibilized reference. Nevertheless, the degradation of both unmodified and IL modified PPS is quite similar with two main steps of degradation identified at 530 °C and 670 °C. 

#### 3.1.2. Confirmation of the Presence of ImCl by FTIR Analysis 

FTIR analysis is a powerful tool to provide information on the presence of ImCl into PPS matrix. According to the literature, the absorbance spectra of PPS is affected by the state of crystallinity of the matrix. In fact, band intensities are subjected to variations caused by increasing the polymer symmetry after crystallization [29]. For this reason, FTIR analyses have been performed in the molten state (300 °C) to compare PPS and PPS_ImCl neglecting crystallization effects on the band intensity. Thus, FTIR spectra of each system i.e., PPS, PPS_ImCl and ImCl are shown in Appendix A. The infrared footprint of ImCl highlighted a main absorption band at 1704 cm^−1^ corresponding to the C=O stretching vibration of uncoordinated carboxylate acids. The signal shows a great correspondence with FTIR analysis of the same ionic compound reported by Ezugwu et al. [30]. Concerning PPS with and without ImCl, similar absorbance peaks are obtained in the molten state. These peak assignments have previously been determined by Zimmerman et al. [29] and are reported in Appendix A. However, the major differences between the two systems are found especially at the maximum absorbance peak of the imidazolium salt. This observation clearly confirms the presence of the ImCl into PPS as previously demonstrated by TGA (Figure 3) and also indicates that the potential reactivity of the salt is preserved after processing at 330 °C with the presence of the C=O stretching at 1704 cm^−1^.

#### 3.1.3. Surface Energies

Contact angle measurements were performed on PPS and PPS_ImCl in order to determine if the incorporation of the imidazolium salt has one influence on the surface energy. Using the Owens–Wendt Raebel and Kabel relation given below, we calculated the non-dispersive, dispersive and total surface energies of PPS and PPS_ImCl.
(3)$$\frac{\gamma_{L}\left(1+\cos\theta \right)}{2{\left(\gamma_{L}^{d}\right)}^{0.5}}=2{\left(\gamma_{S}^{nd}\right)}^{0.5}{\left(\frac{\gamma_{L}^{nd}}{\gamma_{L}^{d}}\right)}^{0.5}+{\left(\gamma_{S}^{d}\right)}^{0.5}$$

The contact angles as well as the polar and the dispersive components of both systems are listed in Table 3 and Table 4. 

It can be seen in Table 4 that the surface energy of PPS is around 46 mN·m^−1^ associated with a low polar component (3.1 mN·m^−1^). Previous studies are in the same range and demonstrate why this polymer has low adhesion on many substrates [31,32]. After the incorporation of 5 wt% of ImCl, an increase of the polar component is observed as well as the total surface energy (51 vs. 46.7 mN·m^−1^). This result can be explained by the presence of two carboxylic acid functions in the imidazolium salt confirming by FTIR measurements. 

#### 3.1.4. Crystallization Behavior of the Matrices

The crystalline morphology and rate play a key role on the micromechanical adhesion measurements which has been underlined in different studies [20,21]. Kobayashi et al. showed the importance of the crystallinity of the matrix on the interfacial shear strength measured by pull-out [20]. In their work, quenched PPS matrices associated with ex-PAN carbon fibers led to poor adhesion measurements while annealing treatment strongly increased the IFSS (from 36 to 71 MPa) [20]. Same observations were reported by Meretz et al. [21] since the range of interfacial shear strength was highly affected by the crystalline state: amorphous PPS disclosed low IFSS with high strength carbon fibers vs. high interfacial resistance after annealing (from 32 MPa to 50–90 MPa).

Considering this fact, the ability of PPS matrix to crystallize before and after the incorporation of ImCl was investigated. First, the thermal history of PPS and PPS containing ImCl were removed. Extrudated samples were kept at 345 °C for 6 min under nitrogen atmosphere to suppress pre-existing crystals found in the molten state. Then, the crystallization temperature and enthalpy, respectively, *T_c_* and Δ*H_c_*, were recorded during a cooling ramp of 10 °C·min^−1^ from 345 °C to 90 °C. The melting temperature, *T_m_*, was obtained during a heating ramp of 10 °C·min^−1^ from 90 °C to 310 °C. The resulting thermograms for PPS and PPS_ImCl are presented in Figure 4 and the summary of the results are listed in Table 5.

The influence of the imidazolium salt induced a significant increase of the crystallization temperature (*T_c_*) from 195 to 232 °C upon the addition of 5 wt% of ImCl (see Figure 4). This result may be due to nucleation effects linked to the presence of the imidazolium salt which acts as heterogeneous nuclei. Another hypothesis could be based on the cleavage of the chain polymer leading to a molecular mass decrease and a better mobility. The modification of the crystalline transformation had previously been observed on PPS blended with liquid crystalline polymer [33,34,35] but also on several polymers compatibilized with ionic liquids [36,37,38]. Usually, imidazolium salts as well as ILs inhibit the crystallization transformation by decreasing the crystallization temperature and crystallization enthalpy. In this case, the crystallization temperature is higher and the final degree of crystallinity of PPS was kept. We calculated the relative transformation *α*(*T*) represented Figure 4 using the following relation:(4)$$\alpha \left(T\right)=\frac{\int_{T_{0}}^{T}\left(\partial H_{c}/\partial T\right)dT}{\int_{T_{0}}^{T_{\infty }}\left(\partial H_{c}/\partial T\right)dT}$$
where *T_0_* and *T_∞_* are, respectively, the beginning and the final crystallization temperature. We observed that the presence of ImCl does not only increase the crystallization temperature but also accelerated the transformation. The transformation curves reflect the difference of speed between the two materials because we observe a greater slope on the PPS_ImCl which states for a faster process.

#### 3.1.5. Crystalline Morphology of PPS Containing ImCl Observed by TEM

Ionic liquids are well-known for playing a structural agent behavior in polymer matrices. For instance, this role has been reported by Lins et al. [39] with the incorporation of ILs into a PVDF matrix. In this study, modifications of the crystalline morphologies depending on the chemical nature of both anions and cations of ILs were observed by TEM. These morphological modifications were accompanied by strong changes of the mechanical properties of the modified PVDF matrices. These previous observations encouraged the use of such characterizations to check if same phenomena could be reported for imidazolium salt modified PPS matrix. Slices of both materials i.e., PPS and PPS_ImCl were observed by TEM and delivered information about the crystalline morphologies (Figure 5).

PPS chains are arranged under form of spherulite structures. The nucleation rate is important since the diameter of these crystalline structures is close to 2 µm. When ImCl is added to the matrix, it modifies the crystalline morphology with a smaller structuration and spherulites are no longer so easily observed. These observations are in accordance with crystallization analysis since high crystallization rates lead to smaller microstructures. Moreover, the distribution of ImCl is not homogeneous and leads to aggregates which do not withstand slicing process. The salt is not dissolved into the PPS matrix but disposed into aggregates of different sizes. 

### 3.2. Study of Three ex-PAN Carbon Fiber Surfaces

#### 3.2.1. SEM Observations of the Carbon Fibers

The three carbon fibers present a classical topology of ex-PAN carbon fibers with surface striations attributed to polymer spinning during their processing, as shown in Figure 6.

The surface of non-treated carbon fibers, NT-CF, also present striations but the surface seems smoother than one on non-sized carbon fibers, NS-CF. Chemical oxidations generally contribute to the development of roughness by corrosion that is why the roughness of NT-CF is well-defined [40]. Sized-carbon fibers, S-CF, appear to be well covered by the epoxy sizing. Even if the traditional secondary electron (SE) detector image indicates that the sizing is applied homogeneously, the SE in lens detector was able to collect high contrast images with discontinuous streaks of sizing on the S-CF (d) surface. The covered surface has been characterized with ImageJ^®^ [41,42] and we determined a percentage of covered surface around 60%. This result first underlines the bad wetting of the sizing but also its inhomogeneous application onto the surface.

#### 3.2.2. Surface Energies of Carbon Fibers

Dynamic contact angle measurements of the carbon fibers were measured with water, ethylene glycol and methylene diiodide in order to determine their surface energies. The contact angles are listed in Table 6. By using the Owens–Wendt Raebel and Kabel relation given in 0, we calculated the non-dispersive and dispersive components of surface energies determined fiber’s surfaces. The results are summarized in Table 7.

The results disclosed the importance of the oxidation process on the total surface energy and more especially on the non-dispersive component which is sensitive to the polar groups present on the carbon fibers. This value increased from 4.2 to 22.1 mN·m^−1^ between NT-CF and NS-CF which clearly demonstrates the appearance of polar groups (oxygen) at the surface of the oxidized carbon fiber. We also observed that the application of the epoxy sizing on the oxidized fiber decreased its polarity. Same observations have previously been reported by Schultz et al. [43]. The authors reported that the polar contribution of the total surface energy followed the same order as **γ^nd^**
_S-CF_ < **γ^nd^**
_NT-CF_ < **γ^nd^**
_NS-CF_. In comparison to the values obtained for PPS and PPS_ImCl, we can estimate that the best interaction should be found with NS-CF because its total surface energy is the highest one promoting a good wettability by the matrix.

#### 3.2.3. Chemical Analysis of Carbon Fiber’s Surfaces

X-ray photoelectron spectroscopy is a powerful quantitative technique which provides information on the elemental composition and also on the chemical functions encountered at the surface of a material (<10 nm). This analysis was employed to better understand the chemical reactivity of the fibers and to confirm the results obtained from wettability analysis. The first information provided by this analysis is the elemental composition of the surfaces (See Table 8).

All surfaces are composed by three major elements: carbon, oxygen and nitrogen which are the usual elements found in ex-Pan high resistance carbon fiber surfaces. Other elements are provided by the chemical oxidation baths during processing or the presence of the sizing. The effect of the different surface treatments i.e., oxidation and sizing application are well highlighted: the oxygen content of the non-treated carbon fiber is relatively low with 3.3 at%, this value increases to 9.5 at% after the chemical oxidation leading to new surface functions. This observation appears to be consistent with the wettability analysis since we previously found a decrease of 30° of the contact angle with water on NS-CF compared to NT-CF. On the over hand, the application of the epoxy layer at the surface is also accompanied by an oxygen content increase which goes from 9.5 at% to 21.6 at%. This increase can be explained by the oxygen content present in the epoxy sizing. High resolution spectra of the carbon pic are shown in Appendix A. The decomposition of the pic C 1s gives access to the proportion of chemical bonds form at the surface. Ninety percent of surface for the NS-CF and NT-CF is composed of aliphatic compounds and/or graphite structure identified with an asymmetric carbon peak at 284.5 eV corresponding to C–C/C–H bonds. These chemical bonds located with a peak at 284.8 eV only mean 60% of the surface for S-CF. The surface bonding of C–O and C–N whose binding energies are located between 286.4–286.8 eV cover 37% of carbon bonds for S-CF whereas they only represent 5% of bonds for NS-CF and are not present on NT-CF. The carboxylic bonds are relatively low on the fibers surface owing to the intensity of peaks corresponding to O–C–O and C=O binding energies located at 287.8 eV and O–C=O bonds located between 288.6–289.3 eV.

### 3.3. Determination of the Matrix-Fiber Adhesion by Micromechanical Tests

Microbond technique was used to determine and compare the interfacial shear strength (IFSS) with both matrices, i.e., PPS and PPS_ImCl as a function of the surface chemistry of three carbon fibers, i.e., NT-CF, NS-CF, S-CF. In this micromechanical test, droplets are debonded via two razors blades (see Figure 7). The force/displacement data was recorded in order to determine the IFSS by considering the homogeneous force along the interface:(5)$$IFSS=\frac{F_{max}}{\pi dl_{e}}$$
where *F_max_* is the debonding force, d the diameter of the carbon fiber and *l_e_* the embedded length measured by optical microscopy.

According to the literature, various micromechanical models have been developed and discussed as in the work of Zinck et al. [44]. However, the majority of these studies are based on an elastic behavior of the matrix during the debonding. This behavior was not observed in our case with modified or unmodified PPS as shown in post-mortem SEM pictures showing a plastic yielding under the blades for both matrices (Figure 8). For this reason, the apparent shear strength was chosen in this study. Meretz et al. have underlined the influence of the embedded length on the apparent shear strength [21]. Their results indicated a strong influence of l_e_ on the interfacial shear strength of crystalline PPS and carbon fibers whereas no influence was got with an amorphous matrix. This has been explained by the difference of shear loading between elastic and ductile matrix. As PPS is semi-crystalline, the determination of the embedded length/surface is a key parameter having an influence on measured IFSS.

From the results presented in Appendix A, we noticed highly scattered IFSS values, especially for S-CF and NS-CF systems, but we observed no noticeable influence of the embedded length on the IFSS. That means that even if our polymers are crystalline, they still possess a ductile behavior as observed via SEM pictures.

#### 3.3.1. Influence of the Chemical Surfaces of Carbon Fibers

The influence of the carbon fiber surface chemistry is noticeable because we found a variation of the IFSS value with the different carbon fibers, as shown in Figure 9 and in Table 9.

The sized (S-CF) and the non-sized carbon fiber (NS-CF) shared a similar adhesion with PPS whereas a lower adhesion is found for the non-treated carbon fiber (NT-CF). These results were expected since we previously underlined the poor reactivity of the NT-CF from XPS measurements. Concerning S-CF and NS-CF, there is no major difference with PPS. This can be explained by the fact that both carbon fibers have been oxidized and have sufficient oxygen bonds to interact with the potential group ends reactive functions of the polymer. However, the addition of the epoxy sizing does not improve the IFSS for S-CF and this can be due to the degradation of the sizing due to the high process temperature of the PPS.

#### 3.3.2. Influence of the Addition of the Imidazolium Salt

The addition of the imidazolium salt into the polymer matrix leads to an increase of the IFSS measured with all carbon surfaces: we observed an increase of 23% on the adhesion with S-CF, 34% with NS-CF and 18% for NT-CF. From the previous characterizations, we can state that two main mechanisms can be responsible for the enhancement of the interphase.

The first one relies on the modification of the polymer crystalline morphology due to the nucleating role of the ImCl. This phenomenon has previously been observed on the work of Trudel et al. at the mesoscale for the adhesion between PPS and CF [19]. The enhancement of short beam strength corresponding to interlaminar shear strength (ILSS) was attributed to the reduction of the spherulites size. On the other hand, the work of Gao et Kim also underlined the influence of the cooling rates on the microstructure as well as the adhesion of PEEK/CF. But in their case, small microstructures leaded to low IFSS and ILSS [45]. In any event, the microstructure has an influence on the adhesion measured at the micro and meso scale and the nucleating effect of the ionic salt on the PPS crystallinity may explain the enhancement of the adhesion.

The second mechanism responsible for the increase of the IFSS may be linked to the reactivity of the ImCl. The reactivity of this imidazolium salt with epoxy groups has previously been investigated by Perchacz et al. [46]. Since we demonstrated the presence of the carboxylic function of ImCl into the PPS matrix able to react with epoxy functions, a possible reaction with sized carbon fibers may occur. However, this reaction may be limited because of the degradation of the sizing at high temperature. Another reaction is possible with the hydroxyl groups. The chemical coupling of the ImCl with the carbon fiber surface must be considered since the adhesion improvement is not significant with the non-treated carbon fiber whereas it remains important for S-CF and NS-CF. Figure 10 displays the potential interactions between the treated carbon fibers and the modified matrix. For the interface with untreated carbon fibers, only π-π interactions may take place between the fiber carbon structure and the aromatic groups present in imidazolium salt and PPS matrix. After oxidation of carbon fiber, additional interactions may take place between the generated polar groups (alcohol, carboxylic groups, etc.) and the reactive groups of imidazolium salt. The sizing adds more reactive groups onto the fiber surface with glycidyl groups able to covalently react with amino group ends of PPS and the carboxylic groups of imidazolium salt. The interface can be the place where more interactions can take place to increase the adhesion.

## 4. Conclusions

In this work, the influence of the surface chemistry of ex-PAN carbon fiber as well as the influence of the addition of an imidazolium salt into the PPS matrix on the adhesion with PPS was studied. The presence of the imidazolium salt in the modified polymer PPS_Imcl was demonstrated by FTIR combined with TGA. Despite the high temperatures employed within the extrusion process, it seems that no major chemical degradations of the salt occurred. Indeed, PPS_ImCl still possesses its acid carboxylic functional groups which ensures a potential reactivity at the interface. We reported that ImCl into PPS did act as heterogeneous nuclei which strongly modified the crystallization and may have an impact of the adhesion. However, the interfacial reinforcement mainly provides from the additional interactions promoted by the ionic salt that migrates towards the interface and supplies both additional π-π interactions and possible reactivity through the reactive carboxylic groups of the imidazolium salt. This thermally stable new compatibilizer is a relevant adhesion promoter able to replace the sizing agent by migration from the matrix towards the fiber. Easy to process by means of conventional extrusion tools, this compatibilization route is simple and efficient to improve the fiber/matrix coupling.

## Data Availability

The data presented in this study are available on request from the corresponding author.

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
