# Peer review of "Imidazolium Salt for Enhanced Interfacial Shear Strength in Polyphenylene Sulfide/Ex-PAN Carbon Fiber Composites"

_polymers, 2022, doi:10.3390/polym14173692_

Round 1
Reviewer 1 Report
In this manuscript, authors have described the use of imidazolium salt (ImCl) for modification of PPS polymer and use of modified PPS polymer for making ex-Pan carbon fiber composites. Authors have clearly defined their goal of increasing interfacial shear strength for thermoplastic PPS/ex-Pan carbon fiber composite, and propose an alternate to thermoset materials. The manuscript includes satisfactory number of experiments to prove their hypothesis, where authors have tested neat PPS along with ImCl modified PPS on three different type of fibers. Authors have performed detailed characterization of the modified PPS polymer and carbon fiber composites made using the PPS polymer. The obtained results are convincing to prove their hypothesis. The manuscript describes the possible mechanisms for increased interfacial shear strength by using ImCl salt and contains satisfactory results to prove those mechanisms.
Overall, the article is well organized, well written and explained, and contains satisfactory citations. I recommend publishing this manuscript in Polymers journal with some minor corrections/answers to questions as mentioned below -
1) Authors have mentioned that they used 5% imidazolium salt, is there any reason behind using this particular amount.
2) In section 1.2.2, microcomposite processing, line number 119, authors have mentioned 'small parts of polymer films', could you please describe in detail about size of the film? Any description about weight or length of the film will be helpful for better understanding of process.
3) For section 1.2.2, actual image of the process/assembly can help in better understanding of processing assembly.
4) For figure 2, Influence of the annealing treatment on the microbond crystalline state, please refer images as (a) and (b), and separate description of each image is necessary in figure description.
5) For line number 142, 146, the heating rates are mentioned in K/min, is it deg oC/min? Please verify it and do appropriate correction if same pattern is followed in rest of the manuscript.
6) For line number 243, authors have mentioned Table 3, it should be Table 4.
7) As mentioned on line number 142, the TGA was performed under air atmosphere, whereas polymer processing was carried out under nitrogen atmosphere. It is worth running TGA under nitrogen atmosphere for ImCl, it might have Td5% above 318 oC.
Author Response
polymers
Manuscript 1866130
Response to Reviewers
Dear Reviewers,
Thank you for your message dated on August 19, 2022 regarding the above referenced manuscript.
Please allow us to first express to you our great appreciation for your attentive reading and meticulous assessment of our manuscript entitled '' Imidazolium salt for enhanced interfacial shear strength in polyphenylene sulfide/ex-Pan carbon fiber composites'' (Manuscript polymers-1866130).
Sincerely, we would like to thank the two reviewers for the very relevant remarks and the complete analysis of the paper. Their reviewing efforts truly provided constructive suggestions and insightful comments that improved the overall quality of the publication.
In the revised manuscript, we have taken into account the constructive remarks to clarify the main objectives of the paper and make the different messages clearer. Besides, an effort was made to answer point-by-point the different reviewers’ comments. These corrections appear in blue color in the revised manuscript.
In agreement with all reviewers, we believe that the paper topic will be attractive to a broad audience that Polymers wishes to reach. Meanwhile, we hope that the revised version has now met the journal publication requirements.
Please find enclosed the answers to the comments from you as follows:
----------------------------------------------------------------------------------------------------------
Reviewer(s)' Comments to Author:
Open Review
(x) I would not like to sign my review report
( ) I would like to sign my review report
English language and style
( ) Extensive editing of English language and style required
( ) Moderate English changes required
(x) English language and style are fine/minor spell check required
( ) I don't feel qualified to judge about the English language and style
|
Yes |
Can be improved |
Must be improved |
Not applicable |
|
|
Does the introduction provide sufficient background and include all relevant references? |
(x) |
( ) |
( ) |
( ) |
|
Are all the cited references relevant to the research? |
(x) |
( ) |
( ) |
( ) |
|
Is the research design appropriate? |
(x) |
( ) |
( ) |
( ) |
|
Are the methods adequately described? |
(x) |
( ) |
( ) |
( ) |
|
Are the results clearly presented? |
(x) |
( ) |
( ) |
( ) |
|
Are the conclusions supported by the results? |
(x) |
( ) |
( ) |
( ) |
Comments and Suggestions for Authors
In this manuscript, authors have described the use of imidazolium salt (ImCl) for modification of PPS polymer and use of modified PPS polymer for making ex-Pan carbon fiber composites. Authors have clearly defined their goal of increasing interfacial shear strength for thermoplastic PPS/ex-Pan carbon fiber composite, and propose an alternate to thermoset materials. The manuscript includes satisfactory number of experiments to prove their hypothesis, where authors have tested neat PPS along with ImCl modified PPS on three different type of fibers. Authors have performed detailed characterization of the modified PPS polymer and carbon fiber composites made using the PPS polymer. The obtained results are convincing to prove their hypothesis. The manuscript describes the possible mechanisms for increased interfacial shear strength by using ImCl salt and contains satisfactory results to prove those mechanisms.
Overall, the article is well organized, well written and explained, and contains satisfactory citations. I recommend publishing this manuscript in Polymers journal with some minor corrections/answers to questions as mentioned below -
- Authors have mentioned that they used 5% imidazolium salt, is there any reason behind using this particular amount.
A small rate of ionic liquid was selected because our expertise in ionic liquid field as additive showed us that a very low content is enough to have a positive impact on the material performance. Between 1 and 5wt% is a relevant rate for modifying the interfaces in a polymer blend or a composite. A higher content of ionic liquid can plasticize the matrix even exsudate. Hence, it is better to avoid it and to work with a low rate.
- In section 1.2.2, microcomposite processing, line number 119, authors have mentioned 'small parts of polymer films', could you please describe in detail about size of the film? Any description about weight or length of the film will be helpful for better understanding of process.
PPS films are prepared by extrusion with a flat die for processing films with a thickness of 200 mm and a length of a few millimeters which is sufficient to cut samples to form a droplet onto carbon fibers.
- For section 1.2.2, actual image of the process/assembly can help in better understanding of processing assembly.
We think that a schematic drawing is more speaking than a real picture on which we can not see the laid droplets onto carbon fiber owing to the drop small size.
- For figure 2, Influence of the annealing treatment on the microbond crystalline state, please refer images as (a) and (b), and separate description of each image is necessary in figure description.
We modified the caption and added the relevant references in the text.
- For line number 142, 146, the heating rates are mentioned in K/min, is it deg oC/min? Please verify it and do appropriate correction if same pattern is followed in rest of the manuscript.
You are right, it is a mistake. We changed all the rates in oC/min.
- For line number 243, authors have mentioned Table 3, it should be Table 4.
The correction is made.
- As mentioned on line number 142, the TGA was performed under air atmosphere, whereas polymer processing was carried out under nitrogen atmosphere. It is worth running TGA under nitrogen atmosphere for ImCl, it might have Td5%above 318 o
Enclosed the TGA under nitrogen atmosphere. As you can seen, the TGA under nitrogen or air atmosphere are very similar.
polymers
Manuscript 1866130
Response to Reviewers
Dear Reviewers,
Thank you for your message dated on August 19, 2022 regarding the above referenced manuscript.
Please allow us to first express to you our great appreciation for your attentive reading and meticulous assessment of our manuscript entitled '' Imidazolium salt for enhanced interfacial shear strength in polyphenylene sulfide/ex-Pan carbon fiber composites'' (Manuscript polymers-1866130).
Sincerely, we would like to thank the two reviewers for the very relevant remarks and the complete analysis of the paper. Their reviewing efforts truly provided constructive suggestions and insightful comments that improved the overall quality of the publication.
In the revised manuscript, we have taken into account the constructive remarks to clarify the main objectives of the paper and make the different messages clearer. Besides, an effort was made to answer point-by-point the different reviewers’ comments. These corrections appear in blue color in the revised manuscript.
In agreement with all reviewers, we believe that the paper topic will be attractive to a broad audience that Polymers wishes to reach. Meanwhile, we hope that the revised version has now met the journal publication requirements.
Please find enclosed the answers to the comments from you as follows:
----------------------------------------------------------------------------------------------------------
Reviewer(s)' Comments to Author:
Open Review
(x) I would not like to sign my review report
( ) I would like to sign my review report
English language and style
( ) Extensive editing of English language and style required
( ) Moderate English changes required
(x) English language and style are fine/minor spell check required
( ) I don't feel qualified to judge about the English language and style
|
Yes |
Can be improved |
Must be improved |
Not applicable |
|
|
Does the introduction provide sufficient background and include all relevant references? |
(x) |
( ) |
( ) |
( ) |
|
Are all the cited references relevant to the research? |
(x) |
( ) |
( ) |
( ) |
|
Is the research design appropriate? |
(x) |
( ) |
( ) |
( ) |
|
Are the methods adequately described? |
(x) |
( ) |
( ) |
( ) |
|
Are the results clearly presented? |
(x) |
( ) |
( ) |
( ) |
|
Are the conclusions supported by the results? |
(x) |
( ) |
( ) |
( ) |
Comments and Suggestions for Authors
In this manuscript, authors have described the use of imidazolium salt (ImCl) for modification of PPS polymer and use of modified PPS polymer for making ex-Pan carbon fiber composites. Authors have clearly defined their goal of increasing interfacial shear strength for thermoplastic PPS/ex-Pan carbon fiber composite, and propose an alternate to thermoset materials. The manuscript includes satisfactory number of experiments to prove their hypothesis, where authors have tested neat PPS along with ImCl modified PPS on three different type of fibers. Authors have performed detailed characterization of the modified PPS polymer and carbon fiber composites made using the PPS polymer. The obtained results are convincing to prove their hypothesis. The manuscript describes the possible mechanisms for increased interfacial shear strength by using ImCl salt and contains satisfactory results to prove those mechanisms.
Overall, the article is well organized, well written and explained, and contains satisfactory citations. I recommend publishing this manuscript in Polymers journal with some minor corrections/answers to questions as mentioned below -
- Authors have mentioned that they used 5% imidazolium salt, is there any reason behind using this particular amount.
A small rate of ionic liquid was selected because our expertise in ionic liquid field as additive showed us that a very low content is enough to have a positive impact on the material performance. Between 1 and 5wt% is a relevant rate for modifying the interfaces in a polymer blend or a composite. A higher content of ionic liquid can plasticize the matrix even exsudate. Hence, it is better to avoid it and to work with a low rate.
- In section 1.2.2, microcomposite processing, line number 119, authors have mentioned 'small parts of polymer films', could you please describe in detail about size of the film? Any description about weight or length of the film will be helpful for better understanding of process.
PPS films are prepared by extrusion with a flat die for processing films with a thickness of 200 mm and a length of a few millimeters which is sufficient to cut samples to form a droplet onto carbon fibers.
- For section 1.2.2, actual image of the process/assembly can help in better understanding of processing assembly.
We think that a schematic drawing is more speaking than a real picture on which we can not see the laid droplets onto carbon fiber owing to the drop small size.
- For figure 2, Influence of the annealing treatment on the microbond crystalline state, please refer images as (a) and (b), and separate description of each image is necessary in figure description.
We modified the caption and added the relevant references in the text.
- For line number 142, 146, the heating rates are mentioned in K/min, is it deg oC/min? Please verify it and do appropriate correction if same pattern is followed in rest of the manuscript.
You are right, it is a mistake. We changed all the rates in oC/min.
- For line number 243, authors have mentioned Table 3, it should be Table 4.
The correction is made.
- As mentioned on line number 142, the TGA was performed under air atmosphere, whereas polymer processing was carried out under nitrogen atmosphere. It is worth running TGA under nitrogen atmosphere for ImCl, it might have Td5%above 318 o
Enclosed the TGA under nitrogen atmosphere. As you can seen, the TGA under nitrogen or air atmosphere are very similar.

Reviewer 2 Report
Comments
This paper studied a method of interfacial shear strength improvement of fibre composites. The outcome of the paper is interesting however, there are several aspects that need to be improved. The reviewer can only recommend for publication if the author satisfactorily address the following major comments in the revised version.
2. The research questions and justification of selecting variable parameters should be highlighted.
3. Which test standards was considered in this study? How many replicate samples were tested in each category?
5. The novelty of the study should be highlighted more clearly at the end of introduction section. How this study is different from the published study in literature?
6. How the outcome of this study will benefit researchers and end users? This need to be highlighted in introduction or end of conclusion.
7. The research on shear strength enhancement is interesting but not fully novel. Therefore, the recent study in this area should be discussed in introduction section to improve the background information. Recently, bond strength enhancement technique was studied in [Ref: Bond behaviour of composite sandwich panel and epoxy polymer matrix: Taguchi design of experiments and theoretical predictions]. Suggest to include this in introduction section with proper citations to improve the background study.
I would be happy to see the revised version to understand how these comments are being addressed.
Author Response
Open Review
( ) I would not like to sign my review report
(x) I would like to sign my review report
English language and style
( ) Extensive editing of English language and style required
( ) Moderate English changes required
(x) English language and style are fine/minor spell check required
( ) I don't feel qualified to judge about the English language and style
|
Yes |
Can be improved |
Must be improved |
Not applicable |
|
|
Does the introduction provide sufficient background and include all relevant references? |
( ) |
(x) |
( ) |
( ) |
|
Are all the cited references relevant to the research? |
( ) |
(x) |
( ) |
( ) |
|
Is the research design appropriate? |
( ) |
(x) |
( ) |
( ) |
|
Are the methods adequately described? |
( ) |
(x) |
( ) |
( ) |
|
Are the results clearly presented? |
( ) |
(x) |
( ) |
( ) |
|
Are the conclusions supported by the results? |
( ) |
(x) |
( ) |
( ) |
Comments and Suggestions for Authors
Comments
This paper studied a method of interfacial shear strength improvement of fibre composites. The outcome of the paper is interesting however, there are several aspects that need to be improved. The reviewer can only recommend for publication if the author satisfactorily address the following major comments in the revised version.
- The research questions and justification of selecting variable parameters should be highlighted.
As underlined by the first reviewer, we justified the rate of ionic liquids added in the text.
- Which test standards was considered in this study? How many replicate samples were tested in each category?
The micromechanical test used , i.e. microdroplet test is not standardized. This test is relevant to only emphasize the key role of the chemistry of the fiber/matrix interface on the interfacial strength. The mechanical characterization of microcomposites by considering one single fiber within a matrix droplet allows to avoid the effect of parameters such as non-well aligned fibers, porosities, fiber/fiber contacts. Previous works showed the microscale characterization on microcomposites is relevant to forecast the macroscopic behavior of multi-fiber composites materials. 40 single tests were performed for each couple fiber/matrix in order to obtain an averaged IFSS.
- The novelty of the study should be highlighted more clearly at the end of introduction section. How this study is different from the published study in literature?
The novelty of this study is the use of the difunctional ionic salt able to be used as compatibilizer that migrates at the interface to behave as an interfacial agent and to bring a positive impact on the interfacial adhesion. This was highlighted in the introduction and conclusion.
- How the outcome of this study will benefit researchers and end users? This need to be highlighted in introduction or end of conclusion.
This pathway will be useful to reinforce the interfacial adhesion fiber/matrix. Two routes can be explored to improve adhesion : i) development of a new sizing but it is very difficult to modify the sizing of fiber supplier because sizing formulation is always confidential ; or ii) addition of small molecules within the matrix to migrate from matrix towards interface. This last route can be made by the compounder of matrix with extrusion conventional tools and seems to be more relevant.
This highlight was added in the introduction and conclusion.
- The research on shear strength enhancement is interesting but not fully novel. Therefore, the recent study in this area should be discussed in introduction section to improve the background information. Recently, bond strength enhancement technique was studied in [Ref: Bond behaviour of composite sandwich panel and epoxy polymer matrix: Taguchi design of experiments and theoretical predictions]. Suggest to include this in introduction section with proper citations to improve the background study.
We added this citation by referring to another method to improve the interfacial strength in composites.

Round 2
Reviewer 2 Report
I have no further comments.